# GENERALIZING MLPs WITH DROPOUTS, BATCH NORMALIZATION, AND SKIP CONNECTIONS

## ABSTRACT

A multilayer perceptron (MLP) is typically made of multiple fully connected layers with nonlinear activation functions. There have been several approaches to make them better (e.g., faster convergence, better convergence limit, etc.). But the researches lack structured ways to test them. We test different MLP architectures by carrying out the experiments on the age and gender datasets. We empirically show that by whitening inputs before every linear layer and adding skip connections, our proposed MLP architecture can result in better performance. Since the whitening process includes dropouts, it can also be used to approximate Bayesian inference. We have open sourced our code, and released models and docker images at `https://github.com/anonymous`.

## 1 INTRODUCTION

### 1.1 MLP AND ITS TRAINING OBJECTIVE

A multilayer perceptron (MLP) is one of the simplest and the oldest artificial neural network architectures. Its origin dates back to 1958 (Rosenblatt, 1958). Its basic building blocks are linear regression and nonlinear activation functions. One can stack them up to create an MLP with $L$ layers (See Equation 1).

$$\hat{\boldsymbol{y}} = f^{(L)}(\boldsymbol{W}^{(L)}, \ldots, f^{(2)}(\boldsymbol{W}^{(2)} f^{(1)}(\boldsymbol{W}^{(1)} \boldsymbol{x})), \ldots) \tag{1}$$

where $\boldsymbol{x}$, $\hat{\boldsymbol{y}}$, $\boldsymbol{W}^{(l)}$, and $f^{(l)}$ are an input vector, an output vector, the weight matrix, and the nonlinear activation function at the $l$th layer, respectively. Nonlinearity is necessary since without it, an MLP will collapse into one matrix. The choice of a nonlinear function is a hyperparameter. ReLU (Nair & Hinton, 2010) is one of the most widely used activation functions.

In a supervised setup, where we have a pair of a data sample $\boldsymbol{x}^{(i)}$ and a label $\boldsymbol{y}^{(i)}$, we aim to reduce the loss (distance) between the model output $\hat{\boldsymbol{y}}^{(i)}$ and the label $\boldsymbol{y}^{(i)}$. The loss, $L_{\boldsymbol{W}^{(1)},\ldots,\boldsymbol{W}^{(L)}}(\hat{\boldsymbol{y}}^{(i)}, \boldsymbol{y}^{(i)})$, is parameterized by the weights of all $L$ layers. If we have $m$ pairs of data samples and labels, $\{(\boldsymbol{x}^{(1)}, \boldsymbol{y}^{(1)}), \ldots, (\boldsymbol{x}^{(m)}, \boldsymbol{y}^{(m)})\}$, the objective is to minimize the expectation of the loss (See Equation 2).

$$\mathbb{E}_{\mathbf{x},\mathbf{y} \sim \hat{p}_{data}} L_{\boldsymbol{W}^{(1)},\ldots,\boldsymbol{W}^{(L)}}(\boldsymbol{x}, \boldsymbol{y}) = \frac{1}{m} \sum_{i=1}^{m} L_{\boldsymbol{W}^{(1)},\ldots,\boldsymbol{W}^{(L)}}(\boldsymbol{x}^{(i)}, \boldsymbol{y}^{(i)}) \tag{2}$$

Since $m$ can be a large number, we typically batch the $m$ data samples into multiple smaller batches and perform stochastic gradient descent to speed up the optimization process and to reduce memory consumption.

Backpropagation (Rumelhart et al., 1986) is a useful tool to perform gradient descent. This allows us to easily differentiate the loss function with respect to the weights of all $L$ layers, taking advantage of the chain rule.

So far, we talked about the basics of MLPs and how to optimize them in a supervised setup. In the past decade, there have been many works to improve deep neural network architectures and their

performances. In the next section, we'll go through some of the works that can be used to generalize MLPs.

## 2 RELATED WORK

### 2.1 DROPOUT

Dropout (Srivastava et al., 2014) was introduced to prevent neural networks from overfitting. Every neuron in layer $l - 1$ has probability $1 - p$ of being present. That is, with the probability of $p$, a neuron is dropped out and the weights that connect to the neurons in layer $l$ will not play a role. $p$ is a hyperparameter whose optimum value can depend on the data, neural network architecture, etc.

Dropout makes the neural network stochastic. During training, this behavior is desired since this effectively trains many neural networks that share most of the weights. Obviously, the weights that are dropped out won't be shared between the sampled neural networks. This stochasticity works as a regularizer which prevents overfitting.

The stochasticity isn't always desired at test time. In order to make it deterministic, the weights are scaled down by multiplying by $p$. That is, every weight that once was dropped out during training will be used at test time, but its magnitude is scaled down to account for the fact that it used to be not present during training.

In Bayesian statistics, the trained neural network(s) can be seen as a posterior predictive distribution. The authors compare the performance between the Monte-Carlo model averaging and the weight scaling. They empirically show that the difference is minimal.

In some situations, the stochasticity of a neural network is desired, especially if we want to estimate the uncertainty of model predictions. Unlike regular neural networks, where the trained weights are deterministic, Bayesian neural networks learn the distributions of weights, which can capture uncertainty (Tishby et al., 1989). At test time, the posterior predictive distribution is used for inference (See Equation 3).

$$p(\mathbf{y} \mid \mathbf{x}, \mathbf{X}, \mathbf{Y}) = \int p(\mathbf{y} \mid \mathbf{x}, \mathbf{X}, \mathbf{Y}, \mathbf{w}) p(\mathbf{w} \mid \mathbf{X}, \mathbf{Y}) d\mathbf{w} \tag{3}$$

where $\mathbf{w} = \{\mathbf{W}^{(i)}\}_{i=1}^{L}$. Equation 3 is intractable. When a regular neural network has dropouts, it's possible to approximate the posterior predictive distribution by performing $T$ stochastic forward passes and averaging the results. This is called *"Monte-Carlo dropout"* (Gal & Ghahramani, 2016). In practice, the difference between the Monte-Carlo model averaging and weight scaling is minimal. However, we can use the variance or entropy of the results of $T$ stochastic forward passes to estimate the model uncertainty.

### 2.2 BATCH NORMALIZATION

Batch normalization was first introduced to address the *internal covariate shift* problem (Ioffe & Szegedy, 2015). Normalization is done simply by subtracting the batch mean from every element in the batch and dividing it by the batch standard deviation. It's empirically shown that batch normalization allows higher learning rates to be used, and thus leading to faster convergence.

### 2.3 WHITENING INPUTS OF NEURAL NETWORKS

Whitening decorrelates and normalizes inputs. Instead of using computationally expensive existing techniques, such as ICA (independent component analysis) (Hyvärinen & Oja, 2000), it was shown that using a batch normalization layer followed by a dropout layer can also achieve whitening (Chen et al., 2019). The authors called this the IC (Independent-Component) layer. They argued that their IC layer can reduce the mutual information and correlation between the neurons by a factor of $p^2$ and $p$, respectively, where $p$ is the dropout probability.

The authors suggested that an IC layer should be placed before a weight layer. They empirically showed that modifying ResNet (He et al., 2016) to include IC layers led to a more stable training process, faster convergence, and better convergence limit.

## 2.4 SKIP CONNECTIONS

Skip connections were first introduced in convolutional neural networks (CNNs) (He et al., 2016). Before then, having deeper CNNs with more layers didn't necessarily lead to better performance, but rather they suffered from the degradation problem. ResNets solved this problem by introducing skip connections. They managed to train deep CNNs, even with 152 layers, which wasn't seen before, and showed great results on some computer vision benchmark datasets.

## 3 METHODOLOGY

### 3.1 MOTIVATION

We want to take advantage of the previous works on whitening neural network inputs (Chen et al., 2019) and skip connections (He et al., 2016), and apply them to MLPs. In each of their original papers, the ideas were tested on CNNs, not on MLPs. However, considering that both a fully connected layer and a convolutional layer are a linear function of $f$ that satisfies Equation 4, and that their main job is feature extraction, we assume that MLPs can also benefit from them.

$$f(\boldsymbol{x} + \boldsymbol{y}) = f(\boldsymbol{x}) + f(\boldsymbol{y}) \quad \text{and} \quad f(a\boldsymbol{x}) = af(\boldsymbol{x}) \tag{4}$$

The biggest difference between a convolutional layer and a fully connected layer is that the former works locally, but the latter works globally.

Also, even the latest neural network architectures, such as the Transformer (Vaswani et al., 2017), don't use IC layers within their fully connected layers. They can potentially benefit from our work.

### 3.2 NEURAL NETWORK ARCHITECTURE

Our MLP closely follows (Chen et al., 2019) and (He et al., 2016). Our MLP is composed of a series of two basic building blocks. One is called the residual block and the other is called the downsample block.

The residual block starts with an IC layer, followed by a fully connected layer and ReLU nonlinearity. This is repeated twice, as it was done in the original ResNet paper (He et al., 2016). The skip connection is added before the last ReLU layer (See Equation 5).

$$\boldsymbol{y} = ReLU(\boldsymbol{W}^{(2)}Dropout(BN(ReLU(\boldsymbol{W}^{(1)}Dropout(BN(\boldsymbol{x}))))) + \boldsymbol{x}) \tag{5}$$

where $\boldsymbol{x}$ is an input vector to the residual block, $BN$ is a batch normalization layer, $Dropout$ is a dropout layer, $ReLU$ is a ReLU nonlinearity, $\boldsymbol{W}_1$ is the first fully connected layer, and $\boldsymbol{W}_2$ is the second fully connected layer. This building preserves the dimension of the input vector. That is, $\boldsymbol{x}, \boldsymbol{y} \in \mathbb{R}^N$ and $\boldsymbol{W}^{(1)}, \boldsymbol{W}^{(2)} \in \mathbb{R}^{N \times N}$. This is done intentionally so that the skip connections can be made easily.

The downsample block also starts with an IC layer, followed by a fully connected layer and ReLU nonlinearity (See Equation 6).

$$\boldsymbol{y} = ReLU(\boldsymbol{W}Dropout(BN(\boldsymbol{x}))) \tag{6}$$

The dimension is halved after the weight matrix $\boldsymbol{W}$. That is, $\boldsymbol{x} \in \mathbb{R}^N$, $\boldsymbol{W} \in \mathbb{R}^{\frac{N}{2} \times N}$, and $\boldsymbol{y} \in \mathbb{R}^{\frac{N}{2}}$. This block reduces the number of features so that the network can condense information which will later be used for classification or regression.

After a series of the two building blocks, one fully connected layer is appended at the end for the regression or classification task. We will only carry out classification tasks in our experiments for

simplicity. Therefore, the softmax function is used for the last nonlinearity and the cross-entropy loss will be used for the loss function in Equation 2.

Although our MLP consists of the functions and layers that already exist, we haven't yet found any works that look exactly like ours. The most similar works were found at (Martinez et al., 2017) and (Siravenha et al., 2019).

(Martinez et al., 2017) used an MLP, whose order is $FC \rightarrow BN \rightarrow ReLU \rightarrow Dropout \rightarrow FC \rightarrow BN \rightarrow ReLU \rightarrow Dropout$, where $FC$ stands for fully-connected. A skip connection is added after the last $Dropout$. Since $ReLU$ is added between $BN$ and $Dropout$, their MLP architecture does not benefit from whitening.

(Siravenha et al., 2019) used an MLP, whose order is $FC \rightarrow BN \rightarrow Dropout \rightarrow FC \rightarrow BN \rightarrow Dropout \rightarrow ReLU$, where $FC$ stands for fully-connected. A skip connection is added before the last $ReLU$. Their block does include $BN \rightarrow Dropout \rightarrow FC$, like our MLP. However, there is no nonlinearity after the $FC$. Without nonlinearity after a fully-connected layer, it won't benefit from potentially more complex decision boundaries.

Since our neural network architecture includes dropouts, we can also take advantage of the Monte-Carlo dropout (Gal & Ghahramani, 2016) to model the prediction uncertainty that a model makes on test examples.

## 4 EXPERIMENTS

### 4.1 AGE AND GENDER DATASETS

To test our method in Section 3.2, we chose the age and gender datasets (Eidinger et al., 2014) (Rothe et al., 2015). The Adience dataset (Eidinger et al., 2014) is a five-fold cross-validation dataset. This dataset comes from Flickr album photos and is the most commonly used dataset for age and gender estimation. You can check the leaderboard at `https://paperswithcode.com/`. The IMDB-WIKI dataset (Rothe et al., 2015) is bigger than Adience. It comes from celebrity images from IMDB and Wikipedia. We'll report the peformance metrics (i.e., the cross entropy loss and accuracy) on the Adience dataset with five-fold cross-validation so that we can compare our results with others. The IMDB-WIKI dataset is only used for hyperparameter tuning and pretraining.

The original images from the Adience dataset have the size of $816 \times 816 \times 3$ pixels. Although CNNs are known to be good at extracting spatial features from such data, the point of our experiment is to test and compare the MLP architectures. Therefore, we transform the images to fixed-size vectors.

We use RetinaFace (Deng et al., 2020) to get the bounding box and the five facial landmarks, with which we can affine-transform the images to arrays of size $112 \times 112 \times 3$ pixels (See Figure 1). Besides reducing the size of the data, this also has a benefit of removing background noise.

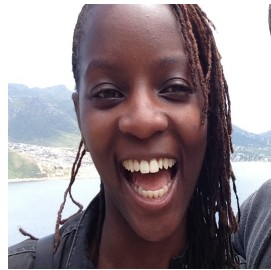 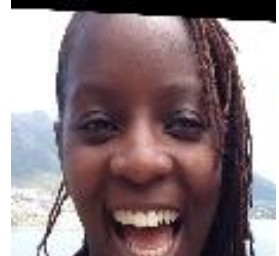

(a) Original image, $816 \times 816 \times 3$ pixels   (b) Affine-transformed to $112 \times 112 \times 3$ pixels

Figure 1: A sample image from the Adience dataset. Instead of using the original images, such as Figure 1a, we use affine-transformed images, as in Figure 1b.

Next, we use ArcFace (Deng et al., 2019) to further reduce the size and extract features from the affine-transformed images. ArcFace is a powerful face-embedding extractor which can be used for face recognition. This transformation transforms $112 \times 112 \times 3$ pixel arrays to 512-dimensional

vectors. It was shown in (Swaminathan et al., 2020) that a classifier can be built on such vectors to classify gender. We take this approach further to even classify age as well.

Table 1 shows the number of data samples before and after removal, along with the gender distribution. Table 2 shows the number of data samples removed and the reason they were removed. Figure 2 shows the age distribution of each dataset. The IMDB-WIKI dataset has a more fine-grained age distribution than the Adience dataset.

| Dataset | number of images before removal | number of images after removal | Gender | |
|---|---|---|---|---|
| | | | female | male |
| Adience | 19,370 | 17,055 (88.04% of original) | 9,103 | 7,952 |
| IMDB-WIKI | 523,051 | 398,251 (76.14% of original) | 163,228 | 235,023 |

Table 1: The statistics of the Adience and IMDB-WIKI datasets

| Dataset | failed to process image | no age found | no gender found | no face detected | bad face quality (det_score <0.9) | more than one face | no face embeddings | total |
|---|---|---|---|---|---|---|---|---|
| Adience | 0 | 748 | 1,170 | 322 | 75 | 0 | 0 | 2,315 (11.95% of original) |
| IMDB-WIKI | 33,109 | 2,471 | 10,938 | 24,515 | 4,283 | 49,457 | 27 | 124,800 (23.86 % of original) |

Table 2: The number of images removed

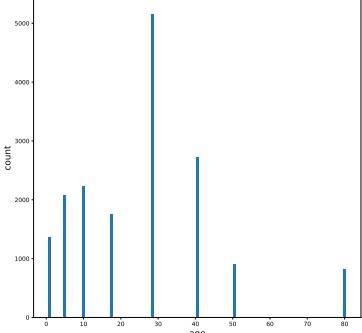

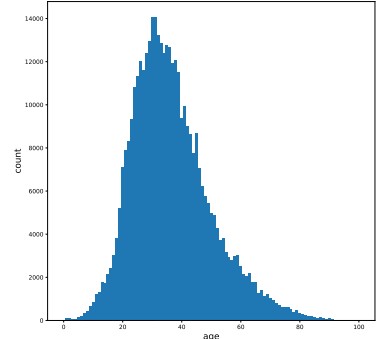

(a) The age distribution of the Adience dataset   (b) The age distribution of the IMDB-WIKI dataset

Figure 2: The age distribution of the Adience (Figure 2a) and IMDB-WIKI (Figure 2b) datasets. Adience has coarse-grained 8 age categories (i.e., 0-2, 4-6, 8-12, 15-20, 25-32, 38-43, 48-53, and 60-100), while IMDB-WIKI has fine-grained 101 age categories (i.e., from 0 to 100).

## 4.2 TRAINING AND HYPERPARAMETERS

We first pretrain three MLPs with the IMDB-WIKI dataset:

1. 2-class gender classification.
2. 8-class age classification[1].
3. 101-class age classification.

There are a number of hyperparameters to be considered to train the models. Some of them are just fixed by following the best practices. For example, we fix the dropout rate to $0.05$ as it was shown in (Chen et al., 2019). We use AdamW (Loshchilov & Hutter, 2019) for the optimizer, as it was proven to be better than using vanilla Adam (Kingma & Ba, 2015). And we use ExponentialLR for learning rate scheduling. Now the remaining hyperparameters to be determined are the number

---

[1]The 101 age categories are converted to the 8 categories by projecting them to the nearest numbers of the Adience dataset.

of residual blocks, number of downsample blocks, batch size, initial learning rate, weight decay coefficient, and multiplicative factor of learning rate decay. They were determined using automatic hyperparameter tuning with Ray Tune (Liaw et al., 2018). We also use automatic mixed precision to speed up training. 10% of the IMDB-WIKI datset was held-out from the other 90% so that weights are backpropagated from the 90% of the data, but the best hyperparameters are determined that have the lowest cross-entropy loss on the 10%.

As for the 2-class gender classification task, the number of residual blocks, number of downsample blocks, batch size, initial learning rate, weight decay coefficient, and multiplicative factor of learning rate decay were chosen to be $4, 4, 512, 2e-2, 4e-3$, and $6e-2$, respectively.

As for the 8-class and 101-class age classification tasks, the number of residual blocks, number of downsample blocks, batch size, initial learning rate, weight decay coefficient, and multiplicative factor of learning rate decay were chosen to be $2, 2, 128, 2e-3, 1e-4$, and $4e-1$, respectively.

Figure 3 shows the neural network architecture of the 2-class gender classification model.

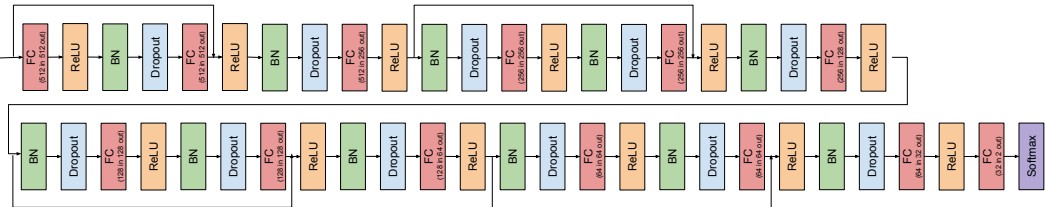

Figure 3: The neural network architecture of the 2-class gender classification model[2]. The skip connections are depicted with the arrows that go over multiple layers.

After the models are trained on the IMDB-WIKI dataset, we fine-tune the models on the Adience dataset. This fine-tuning is running five-fold cross-validation five times to get the least biased results. In this case, the 101-class age classification task should be turned into an 8-class classification task, since the Adience dataset only has 8 age classes. We didn't specifically change the network architecture, since it should be able to learn such a thing by itself by updating its weights.

## 4.3 RESULTS

Table 3 shows the loss and accuracy of all of the trained models. The training stopped if validation loss didn't drop for five consecutive epochs, to reduce training time. All of the models are pretty light-weight. The number of parameters is all between $800,000$ and $900,000$. Training time was very fast. We trained everything on our local machine with an RTX 2060 max-q 6 GB. Automatic hyperparameter tuning, pretraining on the IMDB-WIKI dataset, and five times of five-fold cross-validation on the Adience dataset, took about 1 hour and 10 minutes, 5 minutes, and 13 minutes, respectively.

Although our models were not specifically designed to have the best age and gender classification performance (e.g., A CNN on raw image data probably would have been a better choice), but to test out different MLP architectures, the performance is very competitive, especially for the gender classification task. However, this table is not so easy to interpret. Good performance on the validation split doesn't necessarily lead to good performance on the test split. Also, low cross-entropy loss on the test split somehow doesn't lead to higher accuracy on the test split either.

Therefore, we made the cross-entropy loss vs. epochs plots, in Figure 4. As you can observe, the biggest performance gap between the model with both the IC layer and skip connections and the models with one or both of them removed is displayed in the validation loss with pretrained initialization. The validation loss is consistently lower than the others when the network is initialized with pretrained models. This shows that a network trained on one dataset with IC layers and skip connections helps to generalize to other datasets.

---

[2]We didn't add the IC layer (batch norm and dropout) in the beginning of the neural network. The face embedding vectors themselves were supposed to have their own meanings (e.g., the $L^2$ norm is fixed to 1) and we didn't want to harm them by whitening.

| model | training loss mean (std.) | training accuracy mean (std.) | validation loss mean (std.) | validation accuracy mean (std.) | test loss mean (std.) | test accuracy mean (std.) |
|---|---|---|---|---|---|---|
| 2-class gender classification, random init | 0.0650 (0.0074) | 0.9800 (0.0030) | 0.1180 (0.0150) | 0.9639 (0.0063) | 0.4353 (0.0785) | 0.8406 (0.0273) |
| 2-class gender classification, random init, no dropout | 0.0459 (0.0050) | 0.9874 (0.0020) | **0.1044 (0.0109)** | **0.9672 (0.0039)** | **0.4047 (0.0810)** | **0.8413 (0.0326)** |
| 2-class gender classification, random init, no IC | 0.1879 (0.0618) | 0.9425 (0.0102) | 0.2067 (0.0623) | 0.9336 (0.0121) | 0.4622 (0.0914) | 0.8207 (0.0389) |
| 2-class gender classification, random init, no skip connections | 0.0666 (0.0067) | 0.9805 (0.0021) | 0.1176 (0.0130) | 0.9614 (0.0052) | 0.4217 (0.0865) | 0.8340 (0.0345) |
| 2-class gender classification, random init, no IC, no skip connections | 0.6141 (0.1254) | 0.6119 (0.1568) | 0.6144 (0.1242) | 0.6151 (0.1557) | 0.6581 (0.0581) | 0.5847 (0.1015) |
| 2-class gender classification, pretrained | 0.0332 (0.0021) | 0.9917 (0.0009) | **0.0810 (0.0167)** | **0.9772 (0.0049)** | 0.3120 (0.0604) | 0.8887 (0.0255) |
| 2-class gender classification, pretrained, no dropout | 0.0400 (0.0027) | 0.9899 (0.0013) | 0.0841 (0.0192) | 0.9753 (0.0042) | 0.2768 (0.0457) | 0.8986 (0.0198) |
| 2-class gender classification, pretrained, no IC | 0.0438 (0.0037) | 0.9880 (0.0011) | 0.0895 (0.0149) | 0.9743 (0.0047) | **0.2679 (0.0676)** | **0.9066 (0.0248)** |
| 2-class gender classification, pretrained, no skip connections | 0.1110 (0.0050) | 0.9707 (0.0021) | 0.1389 (0.0128) | 0.9596 (0.0059) | 0.2833 (0.0338) | 0.8937 (0.0171) |
| 2-class gender classification, pretrained, no IC, no skip connections | 0.0633 (0.0033) | 0.9833 (0.0012) | 0.1041 (0.0177) | 0.9693 (0.0061) | 0.2835 (0.0659) | 0.9034 (0.0227) |
| 2-class gender classification SOTA (Hung et al., 2019) | | | | | | *0.8966* |
| 8-class age classification, random init | 0.0147 (0.0075) | 0.9974 (0.0017) | 0.2299 (0.0273) | 0.9342 (0.0064) | 1.6409 (0.1889) | **0.5482 (0.0391)** |
| 8-class age classification, random init, no dropout | 0.0139 (0.0079) | 0.9976 (0.0019) | **0.2259 (0.0303)** | **0.9343 (0.0086)** | 1.6301 (0.1815) | 0.5437 (0.0413) |
| 8-class age classification, random init, no IC | 0.2188 (0.0329) | 0.9288 (0.0138) | 0.4496 (0.0480) | 0.8596 (0.0126) | **1.5591 (0.1445)** | 0.5332 (0.0403) |
| 8-class age classification, random init, no skip connections | 0.0157 (0.0039) | 0.9971 (0.0008) | 0.2330 (0.0261) | 0.9331 (0.0079) | 1.6680 (0.1392) | 0.5395 (0.0374) |
| 8-class age classification, random init, no IC, no skip connections | 0.4202 (0.0797) | 0.8359 (0.0428) | 0.7315 (0.0969) | 0.7548 (0.0431) | 2.0719 (0.2454) | 0.4068 (0.0529) |
| 8-class age classification, pretrained | 0.0442 (0.0111) | 0.9906 (0.0027) | **0.2498 (0.0274)** | **0.9237 (0.0076)** | 1.3567 (0.1020) | **0.6086 (0.0283)** |
| 8-class age classification, pretrained, no dropout | 0.0454 (0.0145) | 0.9906 (0.0036) | 0.2721 (0.0262) | 0.9156 (0.0081) | 1.3589 (0.0774) | 0.5968 (0.0187) |
| 8-class age classification, pretrained, no IC | 0.0904 (0.0202) | 0.9799 (0.0057) | 0.3133 (0.0210) | 0.9013 (0.0082) | **1.3406 (0.0900)** | 0.6032 (0.0308) |
| 8-class age classification, pretrained, no skip connections | 0.0538 (0.0181) | 0.9873 (0.0048) | 0.2599 (0.0226) | 0.9210 (0.0083) | 1.3822 (0.0963) | 0.6047 (0.0270) |
| 8-class age classification, pretrained, no IC, no skip connections | 0.1052 (0.0128) | 0.9751 (0.0037) | 0.3307 (0.0252) | 0.8932 (0.0087) | 1.3568 (0.1311) | 0.5937 (0.0357) |
| modified 8-class age classification, random init | 0.0188 (0.0040) | 0.9968 (0.0008) | 0.2354 (0.0271) | **0.9343 (0.0073)** | 1.6534 (0.1735) | 0.5445 (0.0397) |
| modified 8-class age classification, random init, no dropout | 0.0170 (0.0047) | 0.9973 (0.0009) | 0.2345 (0.0273) | 0.9326 (0.0073) | 1.6319 (0.1586) | **0.5477 (0.0354)** |
| modified 8-class age classification, random init, no IC | 0.4539 (0.0472) | 0.8340 (0.0192) | 0.5811 (0.0408) | 0.7933 (0.0183) | **1.3348 (0.1425)** | 0.5329 (0.0372) |
| modified 8-class age classification, random init, no skip connections | 0.0198 (0.0049) | 0.9965 (0.0009) | **0.2336 (0.0270)** | 0.9326 (0.0081) | 1.6528 (0.1671) | 0.5435 (0.0371) |
| modified 8-class age classification, random init, no IC, no skip connections | 0.8561 (0.1304) | 0.6270 (0.0584) | 0.9393 (0.1106) | 0.5967 (0.0519) | 1.4231 (0.1128) | 0.4270 (0.0388) |
| modified 8-class age classification, pretrained | 0.0455 (0.0116) | 0.9913 (0.0027) | **0.2623 (0.0272)** | **0.9195 (0.0080)** | 1.3636 (0.0694) | 0.6005 (0.0233) |
| modified 8-class age classification, pretrained, no dropout | 0.0444 (0.0127) | 0.9918 (0.0028) | 0.2679 (0.0277) | 0.9196 (0.0078) | 1.4226 (0.1403) | 0.5916 (0.0362) |
| modified 8-class age classification, pretrained, no IC | 0.1100 (0.0216) | 0.9726 (0.0064) | 0.3482 (0.0238) | 0.8930 (0.0065) | 1.3880 (0.0790) | **0.6030 (0.0216)** |
| modified 8-class age classification, pretrained, no skip connections | 0.0450 (0.0078) | 0.9908 (0.0021) | 0.2650 (0.0285) | 0.9188 (0.0080) | 1.4405 (0.0882) | 0.5985 (0.0226) |
| modified 8-class age classification, pretrained, no IC, no skip connections | 0.1311 (0.0221) | 0.9655 (0.0068) | 0.3538 (0.0230) | 0.8890 (0.0065) | 1.3887 (0.1497) | 0.5967 (0.0344) |
| 8-class age gender classification SOTA (Zhang et al., 2020) | | | | | | *0.6747* |

Table 3: The loss and accuracy of the trained models. The "modified" 8-class age classification models are the ones that were initially trained for the 101-class age classification task, but then repurposed to 8-class classification. Random initialization is done with Kaiming He uniform random initialization (He et al., 2015). We ran five-fold cross validation five times. The reported numbers are the averages of the 25 runs. Early stopping was done to save training time. That's why some trainings lasted longer epochs than the others.

As mentioned in 2.1, since our models with the IC layers include dropout, we can use the Monte-Carlo dropout to estimate the prediction uncertainty that a model makes. We ran 512 forward passes and computed the entropy values. Obviously the more forward passes we make, the better the estimate is, since it's sampling from a distribution. The authors in (Gal & Ghahramani, 2016) used 100 forward passes, but since our model is very light-weight, running 512 passes on a CPU was not a problem. See Figure 5 for the faces with low and high entropy. Interestingly, four out of the ten highest entropy faces are either baby or kid faces, which has low frequency in the Adience dataset.

# 5 CONCLUSION

MLPs have a long history in neural networks. Although they are ubiquitously used, a systematic research to generalize them has been missing. By placing an IC (Independent-Component) layer to whiten inputs before every fully-connected layer, followed by ReLU nonlinearity, and adding skip connections, our MLP empirically showed it can have a better convergence limit, when its weights are initialized from a pretrained network. Also, since our MLP architecture includes dropouts, we were able to show that our MLP can estimate the uncertainty of model predictions. In the future work, we want to test our idea on different datasets to see how well it can generalize.

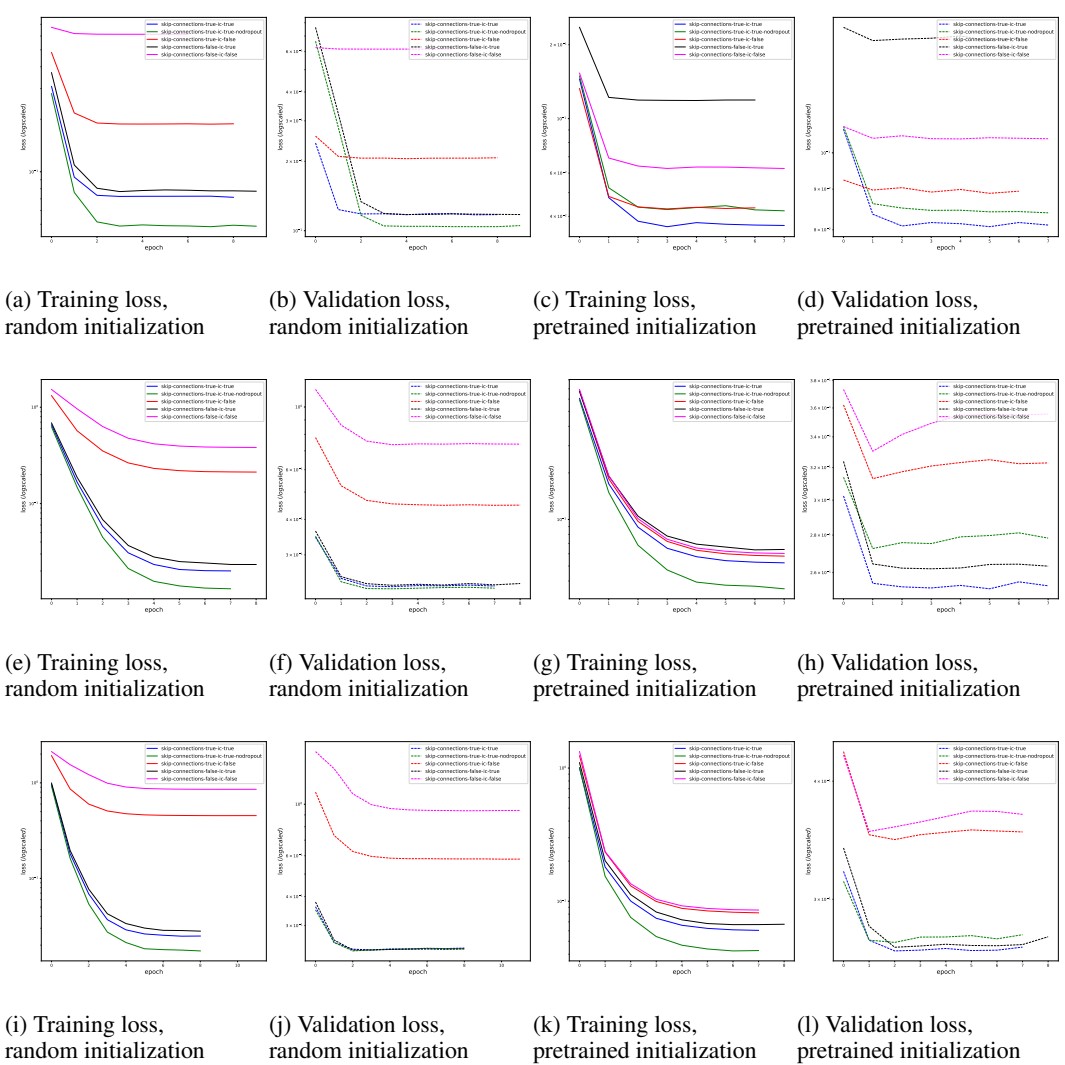

(a) Training loss, random initialization

(b) Validation loss, random initialization

(c) Training loss, pretrained initialization

(d) Validation loss, pretrained initialization

(e) Training loss, random initialization

(f) Validation loss, random initialization

(g) Training loss, pretrained initialization

(h) Validation loss, pretrained initialization

(i) Training loss, random initialization

(j) Validation loss, random initialization

(k) Training loss, pretrained initialization

(l) Validation loss, pretrained initialization

Figure 4: Cross-entropy loss vs. epoch in training gender and classification, with random and pre-trained initialization. The top, middle, and the bottom rows are the 2-class gender, 8-class age, and "modified" 8-class age classification task, respectively. Random initialization is done with Kaiming He uniform random initialization (He et al., 2015). We ran five-fold cross validation five times. The reported numbers are the averages of the 25 runs.

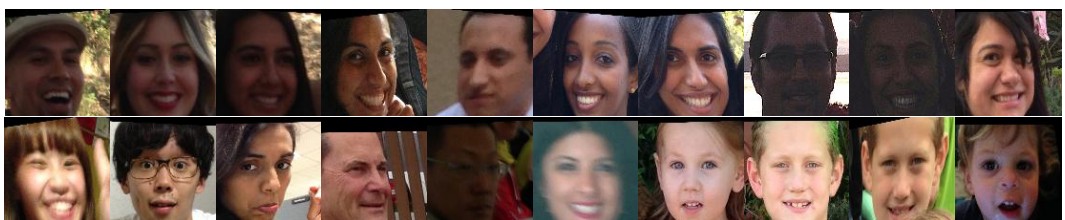

Figure 5: Top row faces are the ten Adience pictures with the lowest entropy. Their entropy values are all zero, which means that the model is $100\%$ sure of its predictions. Bottom row faces are the ten Adience ones with the highest entropy. Their entropy values are all close to $0.6931$, which is the maximum entropy for a binary classification.

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
