# OpenReview forum: "Generalizing MLPs With Dropouts, Batch Normalization, and Skip Connections"
_ICLR.cc/2022/Conference — ICLR 2022 Submitted_

### Official Review · Reviewer_8xR5 · 2021-10-29

**Correctness:** 4
**Technical Novelty And Significance:** 2
**Empirical Novelty And Significance:** 1
**Recommendation:** 3
**Confidence:** 4

**Main Review:**

The authors claim that no-one had proposed the actual block for MLP before. Although several previously published works came close, either the order of layers were slightly different, or they were shown for CNNs. So, potentially, the proposed block architecture could be useful to many practitioners. Unfortunately, the empirical results do not convince one that their particular block is superior. Indeed, only the absence of both skips and whitening is shown to be worse consistently across all tasks. Other combinations offer less clear results, with some winning for some tasks and some other winning for other tasks. Also, the ablation does not include previously published similar blocks (FC-BN-R-D by Martinez et.al. 2017, and FC-BN-D-FC-BN-D-R by Siravenha et.al.2019). Therefore it is not made clear that the particular order of layer in the proposed block is providing any benefits empirically. Furthermore, such claims should be supported across multiple datasets of several types. Here, only 2 similar face datasets are used on 3 similar tasks of age and gender recognitions.


**Summary Of The Paper:**

This paper proposes a general architecture for MLPs. The main ingredients are skip connections and whitening. These methods have been proven to be effective in CNNs. The authors show that MLPs do benefit from them as well. The proposed skip connection (SO,SI - skip-out, skip-in) is added before the last ReLu (R) of each block. The whitening (BND) is implemented by a batch normalization (BN) layer followed by a dropout (D) layer before the fully-connected layer (FC). The proposed block is then: SO-BND-FC-R-BND-FC-SI-R-BND-FCS-R. While FC has the same number of input as output, FCS halves its number of output. The experimental section shows the performance of such MLPs, ablating skips and whitenings. Two datasets of faces are used (IMDB-Wiki and Adience) with 3 classification task of age and gender recognition.


**Summary Of The Review:**

A (somewhat)novel, generic MLP block is proposed. Unfortunately it is not shown to outperform other blocks empirically.

---

### Official Review · Reviewer_qoCZ · 2021-11-02

**Correctness:** 1
**Technical Novelty And Significance:** 2
**Empirical Novelty And Significance:** 2
**Recommendation:** 3
**Confidence:** 4

**Main Review:**

The manuscript itself is clearly written and self-contained.
The idea of the authors builds upon successful deep learning techniques.
Based on these, the authors suggest a new neural network architecture for MLPs.
To our point of view the idea of having residual blocks combined with down-projection layers may be of a certain interest, however from an overall point of view the technical novelties of the paper seem limited.
Since there are no new theoretical analyses, that would justify the proposed network architecture, it would be necessary that the authors provide a convincing experimental evaluation of their approach, which seems to be lacking and which is therefore our major concern with the paper, although it should be mentioned that it seems that the experiments in the publication have been carried out in a technically correct way.
At the end of the paper, the authors provide an analysis on MC Dropout with their architecture, which seems interesting, but which would also require more detailed comparisons to the behavior of MC Dropout with other architectures and a more detailed quantitative analysis w.r.t. data that is infrequent in the training sets.

###### Major concern

The authors mainly only provide an ablation study, but no extensive comparison to other architectures/methods and the results don't look superior to already reported results.
Especially, for age classification their results seem worse than the reported result from Zhang et al. (2020).
The architecture seems to have better results for gender classification (best reported test accuracy in table seems better than Hung et al. (2019)), however based on the validation result, the corresponding architecture would possibly not have been selected and furthermore this best architecture doesn't use relevant novelties of their architecture idea (no IC).
Furthermore, it would be interesting, whether the architecture would also work in an end-to-end way well (e.g., at the top of a CNN), and it would be interesting how well the architecture especially works for tabular datasets as e.g., the Tox21 dataset with circular fingerprints (e.g., ECFP features) or e.g., datasets from the UCI machine learning repository.

**Summary Of The Paper:**

The paper proposes a network architecture for MLP networks.
The authors build upon work by Chen et al. (2019) to achieve whitening by batchnorm followed by dropout (IC Layer).
Further they use skip-connections (He et al., 2016) in their architecture.
The authors tested their approach in an ablation study for gender and age classification.
With dropout included in their architecture, they computed the entropy values, which showed high values
for some baby or kid faces, that were infrequent in the dataset.

**Summary Of The Review:**

Since the experimental evaluation of the proposed network architecture was not convincing for us, and since there are no new theoretical analyses in the publication, we have doubts on the relevance of this paper for ICLR and therefore recommend rejecting the paper.

---

### Official Review · Reviewer_8daM · 2021-11-02

**Correctness:** 2
**Technical Novelty And Significance:** 1
**Empirical Novelty And Significance:** 1
**Recommendation:** 3
**Confidence:** 5

**Main Review:**

The paper is casually written, the 'whitening' is not a real whitening. This kind of basic analysis is of little interest to the community. The authors may investigate some fast whitening methods that are still not widely used in transformers to make the work a little more useful.


**Summary Of The Paper:**

This paper does not bring new knowledge to the community. MLPs have been widely generalized with dropout,BN and skip connections.

**Summary Of The Review:**

The paper is below the ICLR standard.

---

### Official Review · Reviewer_7ffG · 2021-11-03

**Correctness:** 3
**Technical Novelty And Significance:** 2
**Empirical Novelty And Significance:** 2
**Recommendation:** 3
**Confidence:** 4

**Main Review:**

The primary strength of the paper is the careful experimental study the authors conducted to show the utility of the Independent Component layer and skip connections in building deep feedforward neural networks. However, I believe the paper's contributions are marginal.

There are a few suggestions in which the authors may try to build from the current version.

1) Can the authors comment on any significant differences observed in the experimental findings from MLPs and CNNs?

2) What are the reasons behind the improvements in performance with skip connections and the IC layer? Instead of just observing the final accuracy and loss numbers, it will be interesting to see if skip connections and IC layer improve performance because of some underlying phenomenon. Examples of such phenomenons can be smoothening the loss landscape or some underlying implicit bias of skip connections. It will be interesting to track down different function properties learned by the neural networks along the trajectory.

Another question is whether we have faster optimization with the skip connections and the IC layer.

3) The authors can maneuver into studying the properties of the trained network with and without the IC layer and skip connections. E.g., do the trained networks have better adversarial robustness with the IC layer and skip connections.

The authors can build upon many of these questions to deepen their study on the structure of neural networks.

**Summary Of The Paper:**

The authors conduct a careful study on improving the generalization capabilities of multilayer perceptions using dropout, batch normalization, and skip-connections. They show that including an Independent Component layer (using batch normalization and dropout) before every fully connected layer, followed by a ReLU nonlinearity, and adding skip connections, help pretrained MLPs to generalize well on different datasets.

**Summary Of The Review:**

The authors have conducted a careful experimental study to build better MLPs. However, in its current form, I believe the paper's contributions are marginal. However, I think the direction of study is pretty exciting and can be extended to study many related exciting directions.

---

### Decision · Program_Chairs · 2022-01-20

**Decision:**

Reject

**Comment:**

The reviewers unanimously recommend rejecting this submission, and I concur with that recommendation. This submission is not appropriate for a machine learning conference like ICLR. It does not display a thorough understanding of the literature nor does it make a sufficiently valuable contribution. There is no need to "generalize" MLPs, the community knows quite well that we can use dropout, skip connections, and batch norm with them. Even the original dropout paper applies dropout on fully connected ReLU MLPs.

As another example, the submission attributes skip connections to He et al. 2016, but skip connections (also known as "shortcut connections" were in common use in the late 1980s and throughout the 1990s in the Connectionist community, including for non-convolutional simple feedforward neural networks or "MLPs". They were a well known technique throughout neural network history, although the advent of deeper layered neural network architectures perhaps gave them new importance. He et al. certainly popularized them for modern neural network architectures and popularized their residual formulation. The earliest reference I could find easily for skip connections was "Learning to Tell Two Spirals Apart" which was published in 1988 by Kevin J. Lang and Michael. J. Witbrock, but in general such architectural tricks were not viewed as particularly remarkable in the 1990s neural networks literature.